A lightweight fabric defect detection with parallel dilated convolution and dual attention mechanism

Zhang Zheqing 1
Lu Kezhong 1
Yang Gaoming 2 gmyang@aust.edu.cn
1 School of Big Data and Artificial Intelligence, Chizhou University , Chizhou, Anhui , China
2 School of Computer Science and Engineering, Anhui University of Science and Technology , Huainan, Anhui , China
Sergi Consolato
Electronic publication date: 2025 Aug 21
Publication date: 2025
Volume: 11
Electronic Location ID: e3136
Received 2025 Feb 13; Accepted 2025 Jul 29
Copyright: © 2025 Zhang et al.
Copyright year: 2025
Copyright holder: Zhang et al.
License: This is an open access article distributed under the terms of the Creative Commons Attribution License, which permits unrestricted use, distribution, reproduction and adaptation in any medium and for any purpose provided that it is properly attributed. For attribution, the original author(s), title, publication source (PeerJ Computer Science) and either DOI or URL of the article must be cited.
License URL: https://creativecommons.org/licenses/by/4.0/

Keywords: Fabric defect detection, Lightweight model, Parallel dilated convolution, Dual attention

Funding: Natural Science Research Key Project of Department of Education Anhui Province, China 2022AH051828 This work is supported by the Natural Science Research Key Project of Department of Education Anhui Province, China (Grant NO. 2022AH051828). The funders had no role in study design, data collection and analysis, decision to publish, or preparation of the manuscript.

==============================
Detecting defects in fabrics is essential to quality control in the manufacturing process of textile productions. To increase detection efficiency, a variety of automatic fabric defect detections have been developed. However, most of these methods rely on complex model with heavy parameters, leading to high computational costs that hinder their adaptation to real-time detection environments. To overcome these obstacles, we proposed a lightweight fabric defect detection (Light-FDD), building upon the You Only Look Once v8 Nano (YOLOv8n) framework with further optimizations. Specifically, the backbone employed an improved FasterNet architecture for feature extraction. In order to capture multi-scale contextual information, we designed a parallel dilated convolution downsampling (PDCD) block to replace the conventional downsampling block in the backbone. In addition, a novel dual attention mechanism, called the global context and receptive-filed (GCRF) attention, was presented to help the model focus on key regions. Furthermore, a lightweight cross-stage partial (CSP) layer was deployed by dual convolution for feature fusion, reducing redundant parameters to further lighten the model. Results from extensive experiments on public fabric defect datasets showed that Light-FDD outperforms existing state-of-the-art lightweight models in terms of detection accuracy while requiring low computational cost. The present study suggests that the performance and effectiveness of detection models can be balanced through the adoption of reasonable strategies.

Introduction

Fabric surfaces inevitably exhibit discernible defects due to factors such as pollution, damage, and machine failure. Therefore, detecting defects on fabric surfaces is an important procedure for maintaining high-quality standards in fabric production (Hanbay, Talu & Özgüven, 2016). However, traditional fabric defect detection methods, which mainly rely on human workers, are often time-consuming and exhibit low accuracy due to inadequate attention, subjective judgment, and so on (Wu et al., 2019). Hence, it is urgent to develop automatic fabric defect detection systems in the fabric industry.

Several ingenious methods have been presented for fabric defect detection, which can be classified into vanilla methods and deep-learning methods. The former mainly based on image processing techniques, they constructed the detection models from four aspects: statistical (Tsai et al., 2012), spectral (Kumar & Pang, 2002), structural (Shi et al., 2021), and model-based (Tsang, Ngan & Pang, 2016). Although these traditional approaches can yield satisfactory results for certain productions, the complexity and diverse texture features of fabrics make them difficult to generalize in new operational environments. In light of the remarkable success of computer vision algorithms in object detection, the latter have taken over as the mainstream methods in the field of anomaly and defect detection (Shone et al., 2018; Peng et al., 2024; Tang et al., 2024; Yi et al., 2024; Chen et al., 2024).

In recent years, numerous high-performance fabric defect detections based on deep-learning have been designed by researchers. For example, Lu & Huang (2024) proposed a text-aware approach for fabric defect detection by emphasizing the perception of normal fabric textures. Zhang et al. (2023) presented a color conversion detection method that accounts for the impact of fabric color space on detection accuracy. Zahra et al. (2024) localized the defect areas using the saliency-based region technique. Li & Kang (2024) designed a three-stage cascaded network, called Mobile-YOLO, which uses two lightweight networks to first detect the presence of defects and then forwards only the confirmed defect features to a detection network for further analysis. However, feature enhancement by additional structures improves the performance at the cost of increased model complexity. Xiang et al. (2023) introduced an improvement in feature fusion networks for fabric defect detection, effectively aggregating multiscale contextual information while minimizing the increase in model parameters. Although such approaches effectively leverage various levels of features in feature fusion networks without much cost increase, they still overlook the influence of the backbone network on the model complexity. Actually, the backbone network, as the core part of the detection model, was used for processing and extracting meaningful features from the original images. Similar to the feature fusion network, the backbone is also a critical factor that dictates the performance and complexity of detection. Additionally, the decline in detection accuracy caused by the lightweight design should not be overlooked.

In this article, a lightweight fabric defect detection (Light-FDD), building upon the You Only Look Once v8 Nano (YOLOv8n) structure (Jocher, 2024), was proposed. The method employed a strategy in which lightweight design was simultaneously applied to both the backbone and feature fusion network. First, we utilized an improved FasterNet as the backbone for extracting features from the original images. To mitigate the accuracy degradation caused by the lightweight design, we incorporated a parallel dilated convolution as downsampling blocks in the backbone. In addition, a novel channel-spatial attention mechanism, called global context and receptive-field (GCRF) attention, was introduced to enhance the representational capabilities. Furthermore, to reduce redundant parameters of the feature fusion network, lightweight bottleneck and cross-stage partial layers were constructed by using dual convolution operations. These approaches made our model lightweight and more accurate. The main contributions of this article are summarized as follows: (1) Taking into account both the feature fusion network and the backbone network, we proposed a lightweight fabric defect detection model, called Light-FDD. Experimental results demonstrated that our proposed model achieved lower model complexity and faster inference speed while achieving superior detection accuracy compared to state-of-the-art algorithms by applying comprehensive lightweighting strategies throughout the network.

(2) A parallel dilated convolution was designed, and we utilized it to replace vanilla downsampling blocks in the backbone. This approach captures more valuable contextual information, and enhances the interaction between local features and their surroundings by employing multi-dilated convolutions.

(3) A novel dual attention mechanism was incorporated into the detection. This mechanism enhanced the representative ability of the model, which highlights the importance of each spatial feature in the receptive field as well as the long-range dependencies in the global context of each channel.

(4) We utilized dual convolution to construct the feature fusion pyramid structure for light-weight bottleneck and cross-stage partial (CSP) layers. This approach maintains the robustness of the model while reducing its computational complexity.

The remainder of this article is organized as follows: “Related Work” reviews prior works on fabric defect detection. “Methods” describes the details of our proposed model. “Experiment and Results” evaluates the effectiveness and accuracy of our model through a series of comparison experiments. “Conclusions” concludes the article.

Related work

Automatic fabric defect detection

One of the most critical issues in the fabric industry is the detection of fabric defects, which significantly influences the quality of fabric production. The two primary categories of automatic fabric detection techniques are vanilla methods and deep-learning methods. Vanilla methods rely on manually designed pixel-level features to analyze fabric images, including statistical (Tsai et al., 2012), spectral (Kumar & Pang, 2002), structural (Shi et al., 2021), and model-based methods (Tsang, Ngan & Pang, 2016). For example, Tsai et al. (2012) proposed a fast regularity measure for detecting defects on non-textured and homogeneously textured surfaces by analyzing the eigenvalues of covariance matrices formed from pixel gray levels. Kumar & Pang (2002) presented a new supervised defect detection method using Gabor wavelet features for automated textile inspection. Shi et al. (2021) analyzed the difference structure between defect and defect-free samples, and proposed the low-rank decomposition methods based on the high gradient information in defect images. Tsang, Ngan & Pang (2016) considered defective and non-defective regions as the relative skill levels of players in two-player games. They utilized the EIo rating as the predefined computational model to achieve fabric defect detection. Although these traditional approaches can yield satisfactory results, they often struggle with subtle defects and complex textures due to their reliance on handcrafted features and fixed rules.

Due to the great successes of deep learning in image classification and object detection, it has become the main approach to fabric defect detection in recent years. Jing et al. (2017) presented a modified AlexNet for yarn-dyed fabric defect classification. Wei et al. (2019) proposed a combination of compressive sensing and convolutional neural network (CNN) to deal with small sample size problems in fabric defect classification. These investigations demonstrated that deep-learning was a feasible substitute for feature extraction and feature fusion in fabric images. Consequently, a large number of fabric defect detectors based on deep learning have been developed, which can be classified as one-stage and two-stage detectors (Wu, Sahoo & Hoi, 2020).

Since two-stage detectors have higher model complexity than one-stage detectors, they may not meet the requirement of lightweight models. Therefore, this article focuses on one-stage detectors. Lu & Huang (2024) suggested a texture-aware network that incorporates a multi-task defect detection head and an adaptive feature fusion module to improve the generalization of the detection method. Zhang, Qi & Wang (2023) analyzed that the color space of normal fabric textures differs from that of defective regions. This characteristic may significantly improve the performance of fabric defect detection. Hence, they projected the color space from RGB into an optimized space for their detection model. Xiang et al. (2023) proposed a hook-shaped feature pyramid network to aggregate various levels of features for the purpose of detecting fabric defects. The feature map of each level was enhanced by a parallel dilated attention module, which facilitates better interaction between local and global channels. However, these methods only focus on improving the model detection accuracy, neglecting that detection efficiency is also a key indicator in fabric defect detection. Therefore, on the basis of one-stage detection method, we proposed a detection model that not only as higher accuracy but also lower computational costs in this article.

Lightweight for neural networks

To satisfy the requirements of real-time tasks, researchers and practitioners are dedicated to designing lightweight and cost-effective fast neural networks with low latency. One of the general strategies is designing an efficient convolutional filter. Szegedy et al. (2015) demonstrated the effectiveness of using smaller convolutional kernels to replace larger ones, which reduces the computational complexity while preserving the ability to capture spatial dependencies. Howard et al. (2017) employed depthwise separable convolutions to design an efficient deep-learning model. Zhang et al. (2018) utilized group convolution as the filter operation to reduce the computational cost in their model. Singh et al. (2019) introduced heterogeneous convolution (HetConv), which employed kernels of varying sizes within a convolution filter. They employed a few kernels of large sizes (e.g., 3 × 3) and others of small sizes (e.g., 1 × 1) to reduce the computational complexity of the model. Zhong, Chen & Mian (2022) introduced dual convolution, which enhances representational capacity while preserving computational efficiency, by combining the advantages of group convolution (GroupConv) and HetConv. However, simply reducing the number of parameters and floating-point operations (FLOPs) in deep neural networks cannot guarantee improved inference speed due to the frequent memory access caused by fragmented operations.

Han et al. (2020) argued that many feature maps in neural networks are redundant or highly similar, leading to unnecessary computations. To address this, they presented a method of generating “ghost” feature maps from a smaller number of original feature maps. Li et al. (2021) decomposed and integrated sparse connections to reduce FLOPs. Chen et al. (2023) proposed an ultra-lightweight model called FasterNet, designed to reduce both model complexity and memory access overhead. This is achieved through the use of partial convolution (PConv), a novel convolution operator that applies standard convolution to only a portion of the input channels, while preserving the original information in the remaining channels. Considering that both the backbone and feature fusion networks are key parts of the detection model, they significantly impact the overall computational complexity. We selected the main structure of FasterNet as the backbone for our approach, and improved it by several efficient strategies. Furthermore, we employed efficient convolutional filters, specifically dual convolution, to design a lightweight CSP layer and bottleneck in feature fusion networks, yielding satisfactory results.

Methods

This section describes the detailed model in terms of five aspects: the overall structure of Light-FDD, the parallel dilated convolution downsampling (PDCD) block, the global context and receptive-filed attention (GCRF) mechanism, and the lightweight cross-stage partial layers.

Overall structure of Light-FDD

As depicted in Fig. 1, the Light-FDD is composed of three primary parts: a backbone for feature extraction, a path aggregation feature pyramid network (PAFPN) for feature fusion, and a detection head for generating final results. In our model, we utilized an improved FasterNet as the backbone, incorporating two main refined modules. One is the PDCD block, the other is the GCRF attention. Furthermore, we designed a lightweight CSP layer for PAFPN using dual convolution for feature fusion, which improves the model accuracy with lower computational complexity.

Figure 1 The structure of Light-FDD.

First, we improved the performance of the feature extractor by incorporating an enhancement downsampling block and a novel dual attention mechanism in a lightweight backbone. As an ultra-lightweight deep neural network, FasterNet was used as the main structure of our backbone in Light-FDD. Since there are high similarities between feature maps across different channels, FasterNet takes redundant computations into account for vanilla convolutions. To make the detection as lightweight as possible, FasterNet designed partial convolution (PConv), which utilizes only a portion of the input channels in the convolutional operation, leaving the remaining channels unaffected. Additionally, to overcome accuracy reduction brought by lightweight strategies, we modified the downsampling block by the PDCD to capture more features of irregular defects and designed a novel channel and spatial attention mechanism, GCRF attention, enabling the detection to focus on key regions. The architecture of backbone is depicted in Fig. 2.

Figure 2 Backbone network architecture in Light-FDD.

PDCD is the parallel dilated convolution downsampling block, and GCRF is the global context and receptive-field attention mechanism.

Second, we introduced a lightweight design for CSP layers in PAFPN to realize a lower model complexity while having a higher detection accuracy. PAFPN is the main structure in YOLOv8n for feature fusion. It aggregates multi-scale features from various layers of the backbone network and passes them through additional paths to facilitate better information flow. This enables it to capture richer feature representations in the detection model. PAFPN consists of two components: the feature pyramid network (FPN) (Xie et al., 2023) and the path aggregation network (PAN) (Hu et al., 2022). In FPN, information flows predominantly in a top-down manner, from deeper, more abstract features to more localized ones. In contrast, PAN transmits information in a bottom-up path, reusing multi-level feature maps by passing them back through the layers. This enhances feature propagation and improves the details in the feature maps. Notably, the combination of CSP and PAFPN in YOLOv8n significantly boosts the detection accuracy of small object detection. Therefore, we designed a lightweight CSP layer in PAFPN, named “VoVdualCSP”, which reduces computational overhead while preserving high-quality feature representations across different scales.

Finally, the head network of Light-FDD adopted the decoupled-head structure (Qiu, Huang & Tang, 2023), which is commonly used in object detection models to separate the tasks of classification and localization, allowing each task to be independently optimized. This reduces the risk of interference between the two tasks during training, allowing the network to specialize in each aspect of detection. In this article, three outputs from PAFPN are fed into these two detection heads to generate the final results.

Parallel dilated convolution downsampling block

Despite the difficulty of defect detection caused by the uncertain distribution of defect sizes and positions, repetitive textures and patterns are present in fabric images. We can view these characteristics as contextual information, which characterizes the relationship between the object’s semantics and the surrounding scene, providing valuable features for detection to enhance the quality of predictions (Wu et al., 2020; Chalavadi et al., 2022). Therefore, we can make full use of both local details and contextual information to improve detection accuracy. To better extract contextual information over large distances for the discrimination of defects against complex backgrounds, we proposed a novel parallel dilated convolution downsampling (PDCD) block. In this block, two additional dilated convolutions are used to further utilize the contextual information, and the learned features better reflect the texture information of the fabric.

As depicted in Fig. 3, the processing flow of the PDCD block comprises three main stages. First, we employ a standard 3 × 3 convolution with a stride 2 to downsample the input feature map. The reason for this initial downsampling is that subsequent operations involve a large amount of computation when extracting contextual information from fabric images. Thus, we can significantly reduce the computational cost by first downsampling the feature map and its better conforms the design needs of lightweight models. We can formulate this computation process as follows:

(1) Fhalf=Convk=23×3(Fin)

where Fin is the input feature map, Conv represents the standard convolution, and Fhalf denotes the downsampled feature map.

Figure 3 The processing flow of the PDCD block.

The second stage is the parallel dilated convolution module, which is the core of PDCD. Considering the repetitive textures and patterns of fabric images, we designed three dilated convolutions with varying dilation rates to extract different features. Generally, the smallest dilation rate convolution can be regarded as a local feature extractor, which preserves fine-grained details of the feature map. The larger dilation rate serves as a surrounding feature extractor, which extracts features from the surrounding areas of the key regions. The largest dilation rate is used as the texture feature extractor, which focuses on the inherent texture features of fabric images. We uniformly regard the features of the latter two extractors as contextual information. In this article, we set the dilation rates as 1, 3, and 5, respectively. After extracting three different feature maps, we then concatenate them and form a new feature map through the batch normalization (BN), parametric rectified linear unit (PReLU) and 1 × 1 standard convolution operations. The computational process can be represented as follows:

(2) FPD=Convs=11×1(PReLU(BN(Cat(Convs=2,d=13×3(Fhalf),Convs=2,d=33×3(Fhalf),Convs=2,d=53×3(Fhalf))))

where s is stride of convolution, d is the dilation rate.

Finally, to further enhance the representation capability of the feature map, we employed a method similar to the squeeze and excitation (SE) (Hu, Shen & Sun, 2018) attention mechanism to acquire global context information. Specifically, we used global average pooling (GAP) to extract global context features, and utilized two fully connected networks to obtain channel attention weights for recalibrating the feature map by channel-wise multiplicative scaling, which adjusts the importance of each channel in the feature map. This processing procedure can be represented as follows:

(3) Fout=FN(FN(GAP(FPD))⊙FPD

where FN denotes the full connection network, GAP represents global average pooling, and ⊙ is the element-wise multiplication.

Global context and receptive-field attention mechanism

The visual attention mechanism enhances network performance by guiding detection models to focus on important regions. Hu et al. (2018) introduced squeeze-and-excitation (SE) blocks to recalibrate channel-wise features, thereby improving the representational capability of neural networks. Woo, Park & Lee (2018) emphasized the significance of spatial attention and proposed the convolutional block attention module (CBAM), achieving promising results. As evidenced by a large number of studies in the field of image recognition, channel and spatial attention have been shown to be effective in enhancing the model performance. However, there are two main issues with current approaches. On one hand, simply adopting GAP will ignore the long-range dependencies of global context in channel attention, which is beneficial for the tasks of image recognition (Cao et al., 2019). On the other hand, spatial attention does not fully address the parameter sharing problem, which limits the performance of networks (Zhang et al., 2023). To address these limitations, we designed a channel-spatial attention mechanism called global context and receptive-field (GCRF) attention in this article.

As indicated in Fig. 4, the GCRF attention mechanism consists of two blocks, which are the global context block for capturing dependencies among each channel and the receptive-field attention block for emphasizing specific regions of the input feature maps. Specifically, to reduce the compute complexity of the global context in channel attention, we adopt a query independent formulation to compute the global attention map, which enables the detection to emphasize important channels with global context regardless of their spatial positions. In spatial attention, we utilized the concept of receptive-field features to solve the parameter sharing problem completely. In this way, the network can emphasize the significance of spatial features within a receptive-field window and selectively focus on relevant spatial information. The integration of these attention mechanisms enhances performance by enabling the model to selectively emphasize both crucial channel-wise and spatial features, thereby improving its capacity to capture pertinent information and refine its feature representations.

Figure 4 The structure of GC RF attention.

The implementation details of the GC block are depicted in Fig. 5A. It includes two parts: Context Modeling and Transform. For the sake of simplicity, we represented the input feature map as U={ui}i=1H×W, where ui denotes the value of position i, H and W represent the height and width of the feature map, respectively. First, the feature map is processed through a GAP structure, which learns the global representation of each channel from the input feature map. This structure computes global weights Wk for each position using a set of 1 × 1 vanilla convolutions, then applies them to the input feature map by a Softmax function, ultimately yielding the global attention pooling value Zj. The calculation process can be expressed as follows:

(4) Wk=Convs=11×1(U)

(5) Zj=exp(Wkuj)∑m=1H×Wexp(Wkum).

Figure 5 (A and B) Implementation flow of attention mechanism.

Subsequently, we addressed the bottleneck transform through a linear transform matrix Wv. The computation of Wv can be divided into two sequential stages Wv1 and Wv2, which are both obtained by a 1 × 1 vanilla convolution. It should be noted that before computing Wv2, we should perform layer normalization and ReLU activation on Wv1. The GC block output can be expressed by the following equation:

(6) ui′=xi+Wv2ReLU(LN(Wv1∑j=1H×WZjuj)).

The implementation of RF attention is depicted in Fig. 5B. It consists of receptive-filed feature map extraction and feature attention map acquisition modules. Let the input feature map be denoted as U ∈ RC×H×W, where C represents the number of input channels, H and W denote the height and width of the feature map, respectively. The group convolution was used to extract the receptive-field feature map, its kernel size and group number were set to 3 and C × 32. The computational process can be represented as follows:

(7) F1=ReLU(BN(gC×323×3(U))

where BN() indicates the batch normalization, ReLU() refers to the ReLU activation functions, and F1 ∈ R(C×3×3)×H×W denotes the obtained feature map. To achieve feature attention map while minimizing computational overhead, the global information of F1 was first aggregated by AvgPool and MaxPool. Subsequently, a standard 3 × 3 convolution was used to facilitate information interaction. Finally, the Softmax function was utilized to emphasize the significance of each feature. The feature attention map can be computed as follow:

(8) A1=Softmax(f3×3Concat(AvgPool(F1),MaxPool(F1)))

where Concat() is the concatenation operation, f3 × 3 indicates the 3 × 3 standard convolution. The final results of RF attention can be denoted as:

(9) F=fs=k3×3(F1×A1)

where fs=k3×3(⋅) represents the standard 3 × 3 convolution, and the stride is set to equal the kernel size. In this article, we set them both equal to 3.

Lightweight cross-stage partial layer module

In our proposed Light-FDD, we utilized the PAFPN to integrate feature maps from different levels. The main module in PAFPN is the cross stage partial layer. However, although the original CSP Layer of YOLOv8n has achieved satisfactory results, it still has shortcomings in terms of lightweight design. Inspired by Slim-neck (Li et al., 2024), we implemented dual bottleneck by dual convolution (Zhong, Chen & Mian, 2022), and then used it to design a lightweight CSP Layer, VoVDualCSP. As shown in Fig. 6A, dual bottleneck is composed of two dual convolutions and a 3 × 3 standard convolution. Based on the dual bottleneck, we designed the VoVDualCSP layer by employing a one-shot aggregation approach. The structure of VoVDualCSP is indicated in Fig. 6B.

Figure 6 The structure of dual bottleneck and VoVDualCSP layer.

Specifically, the key lightweight approach in the VoVDualCSP layer is the replacement of standard convolution operations with dual convolutions (DualConv). DualConv is a novel lightweight convolutional operator. It incorporates the advantages of group convolution (Krizhevsky, Sutskever & Hinton, 2017) and heterogeneous convolution (Singh et al., 2019). To better explain the computational cost of DualConv, we analyze it using a 3 × 3 convolution kernel. The detailed implementation is illustrated in Fig. 7. DualConv divides the convolutional filters into G groups. Only M/G convolutional kernels perform both 3 × 3 and 1 × 1 convolution, while the remaining (M–M/G) kernels perform only 1 × 1 convolution. In addition, the 3 × 3 convolution shifts across each group while the 1 × 1 convolution operates on all input channels. This manner not only maintains the fundamental characteristics of group convolution but also enhances information interaction and sharing between convolutional layers.

Figure 7 The details of dual convolution.

Purple cuboid denotes 1 × 1 convolution; Blue cuboid denotes 3 × 3 convolution.

Let M denote the number of the input channels, N represent the number of output channels, and k × k be the kernel size of convolutional filters. The number of parameters in standard convolutional layer can be computed as follows:

(10) Para_norm=k×k×M×N.

However, the proportion of k × k convolutional kernels is controlled by the number of groups G in DualConv. The number of parameters in convolutional kernel both performing is k × k and 1 × 1 convolutions is (k × k + 1) × M × N/G, and the number of parameters in the remaining 1 × 1 convolutional kernel is (M-M/G) × N. The total number can be represented as.

(11) Para_dual=(k×k+1)×M×N/G+(M−M/G)×N=k2×M×N/G+M×N.

By Eqs. (10) and (11), DualConv is 5~7 times faster than the standard convolution when k = 3, G = 16. Hence, the VoVDualCSP Layer is more suitable for lightweight models.

Experiment and results

In this section, we evaluated the proposed Light-FDD on a publicly available fabric defect dataset containing four types of defects on fabric surfaces. We conducted several comparative experiments with other lightweight state-of-the-art detection models and performed ablation studies to assess the contribution of each improvement in Light-FDD. Moreover, to further verify the robustness and general applicability of our model, we carried out Light-FDD on another more complex fabric defect detection dataset and achieved the best performance.

Experimental environment

Datasets

In this article, all experiments were conducted using the fabric defec-zqt9u dataset (ak03, 2023), obtained from the Roboflow platform. The dataset includes four types of defects: holes, knots, lines, and stains. It comprises 4,648 images, each with a resolution of 640 × 640 pixels. The dataset was split into training and testing sets with an 8:2 ratio. The detailed information of our dataset is visualized in Fig. 8. Since the distribution of each defect is relatively balanced, no additional data augmentation techniques were applied in this study.

Figure 8 Numbers of each fabric defect type.

Experimental setting

To validate the competitiveness of the proposed Light-FDD, several state-of-the-art object detection models were selected for comparison. For a fair comparison, all detection models were implemented by PyCharm with PyTorch 2.3.1 and Python 3.9.19, and evaluated on the same training, validating and testing sets. All comparative models, including our proposed method, adopted the same data augmentation strategies as the default of YOLOv8 training pipeline: Mosaic, Mixup, and affine transformation. Meanwhile, other default tricks in YOLOv8n including anchor-free bounding-box prediction, adaptive multi-scale input resizing, automatic mixed-precision training, a brief learning-rate, warm-up phase, and a composite bounding-box regression loss (Complete intersection over union (IoU) and distributional focal loss) were used to optimize network training. The parameters for these models were set according to the specifications in the corresponding literature, specifically starting from scratch with the SGD optimizer, weight decay of 0.0005, and momentum of 0.937. The batch size was set to 16, the maximum number of epochs to 300, and the learning rate to 0.01. All experiments were carried out on a workstation equipped with an NVIDIA GeForce RTX 4090 graphics card, with running memory 24 GB, an Intel® Xeon® Gold 5318 @ 2.10 GHz CPU, and 64 GB of memory on Ubuntu 18.04.1 LTS.

Evaluation metrics

In this article, we evaluated the detection performance of our proposed model using several metrics, including precision (P), recall (R), F1 score, mean average precision (mAP), model size, floating-point operations (FLOPs), and frames per second (FPS). The mAP represents the average of the average precision (AP) values across all classes, where AP is calculated as the area under the precision-recall curve. It serves as an indicator of prediction accuracy for each defect type. The model complexity is directly influenced by the model size and FLOPs. In addition, FPS is utilized to assess the inference speed of the model, with a higher FPS corresponding to an increased rate of image processing. The calculations for these metrics are as follows:

(12) P=TPTP+FP

(13) R=TPTP+FN

(14) F1=2×P×RP+R

(15) AP=∫0RP(r)dr

(16) mAP=∑n=1nAPin

where TP (true positive), FP (false positive), and FN (false negative) refer to the number of correctly predicted positive samples, the number of negative samples incorrectly predicted as positive, and the number of positive samples incorrectly predicted as negative, respectively, n implies the number of categories.

Comparison with other methods

To provide a comprehensive analysis of our proposed model, several prominent lightweight models were evaluated on the fabric defec-zqt9u dataset in this article. The main metrics for these detectors are reported in Tables 1 through 5, where bold numbers indicate the best values and underlined numbers denote the second-best ones.

Table 1 Comparison of model complexity among different detectors.

Model	Model size (MB)	GFLOPs	FPS	
ShuffleNetv2	5.7	7.4	103.2	
GhostNetv2	12.7	8.7	45.9	
MobileNetv3-S	4.6	5.5	98.6	
MobileNetv4-S	8.5	8.0	126.5	
YOLOv5n	8.6	7.2	125.2	
YOLOv8n	8.4	8.1	130.4	
YOLOv10n	5.8	8.2	128.9	
YOLOv11n	5.5	6.3	138.7	
Light-FDD (ours)	5.5	6.1	139.5	
Note:

Bold values indicate the best values and underlined values denote the second-best ones.

Table 2 The mean average precision (mAP50) of different detectors.

Model	Type of defects (mAP50)	All (mAP50)	
Hole	Knot	Line	Stain	
ShuffleNetv2	0.887	0.918	0.863	0.679	0.83	
GhostNetv2	0.884	0.911	0.844	0.654	0.823	
MobileNetv3-S	0.853	0.911	0.825	0.607	0.799	
MobileNetv4-S	0.874	0.87	0.843	0.665	0.813	
YOLOv5n	0.896	0.881	0.797	0.704	0.819	
YOLOv8n	0.864	0.894	0.828	0.712	0.824	
YOLOv10n	0.86	0.891	0.794	0.694	0.810	
YOLOv11n	0.889	0.881	0.864	0.71	0.836	
Light-FDD (ours)	0.90	0.897	0.864	0.730	0.848	
Note:

Bold values indicate the best values and underlined values denote the second-best ones.

Table 3 The F1 score values of different detectors.

Models	Type of defects (F1)	All(F1)	
Hole	Knot	Line	Stain	
ShuffleNetv2	0.869	0.874	0.793	0.707	0.811	
GhostNetv2	0.867	0.872	0.763	0.692	0.799	
MobileNetv3-S	0.846	0.886	0.816	0.661	0.802	
MobileNetv4-S	0.851	0.850	0.803	0.693	0.799	
YOLOv5n	0.864	0.871	0.779	0.713	0.807	
YOLOv8n	0.850	0.878	0.774	0.717	0.805	
YOLOv10n	0.843	0.858	0.758	0.691	0.788	
YOLOv11n	0.834	0.851	0.763	0.717	0.791	
Light-FDD (ours)	0.868	0.862	0.834	0.753	0.829	
Note:

Bold values indicate the best values and underlined values denote the second-best ones.

Table 4 Ablation experiment results for each defect category.

Category	Indicator	A	A + B	A + B + C	A + B + C + D	A + B + C + D + E	
Hole	Precision	0.9	0.888	0.884	0.916	0.905	
Recall	0.806	0.839	0.826	0.828	0.838	
mAP50	0.864	0.88	0.883	0.895	0.9	
mAP50-95	0.521	0.513	0.550	0.556	0.573	
Knot	Precision	0.909	0.87	0.888	0.885	0.901	
Recall	0.849	0.794	0.82	0.837	0.827	
mAP50	0.894	0.854	0.876	0.886	0.897	
mAP50-95	0.49	0.418	0.453	0.470	0.480	
Line	Precision	0.779	0.708	0.826	0.851	0.829	
Recall	0.77	0.733	0.823	0.827	0.802	
mAP50	0.828	0.747	0.871	0.872	0.864	
mAP50-95	0.444	0.367	0.475	0.503	0.497	
Stain	Precision	0.853	0.867	0.872	0.863	0.876	
Recall	0.618	0.554	0.621	0.629	0.660	
mAP50	0.712	0.676	0.709	0.714	0.730	
mAP50-95	0.427	0.393	0.435	0.436	0.437	
Note:

Bold values indicate the best values and underlined values denote the second-best ones.

Table 5 Overall ablation experiment results.

Indicator	A	A + B	A + B + C	A + B + C + D	A + B + C + D + E	
Precision	0.86	0.833	0.868	0.879	0.878	
Recall	0.761	0.73	0.772	0.780	0.782	
mAP50	0.824	0.789	0.835	0.842	0.848	
mAP50-95	0.471	0.423	0.48	0.491	0.497	
Model size (MB)	6.4	4.6	5.7	5.9	5.5	
FLOPs(G)	8.1	6.0	6.3	6.5	6.1	
FPS	130.4	194.8	143.3	135.1	139.5	
Note:

Bold values indicate the best values and underlined values denote the second-best ones.

As depicted in Table 1, Light-FDD achieves lower complexity than most advanced lightweight models in terms of model size, FLOPs, and FPS, such as ShuffleNetv2 (Ma et al., 2018), GhostNetv2 (Tang et al., 2022), MobileNetv4-S (Qin et al., 2024), YOLOv5n (Jocher, 2020), YOLOv8n (Jocher, 2024), and YOLOv10n (Wang et al., 2024). Specifically, the model size of Light-FDD is 96.49%, 43.31%, 64.71%, 63.95%, 65.48%, and 94.83% of these models, respectively. The gigaFLOPS (GFLOP) are 82.43%, 70.11%, 76.25%, 84.72%, 75.31%, and 74.39% of theirs, respectively. In addition, the inference time of our proposed model is also faster than the above models. Although our detection has a 0.8 M larger model size and 0.6G higher GFLOPs than MobileNetV3 -S (Howard et al., 2019), its inference speed is 40.9 FPS faster. Notably, the model complexity metrics of YOLOv11n (Khanam & Hussain, 2024) and Light-FDD are similar, while our proposed model still has a slight advantage in terms of GFLOPs and FPS. Thus, from the perspective of model lightweighting, Light-FDD has lower complexity and faster inference speed than other detections. This indicates that our detection offers a lightweight and efficient solution for fabric defect detection.

In addition to lightweight metrics, we further take the detection accuracy into account as the key factor for our Light-FDD. Therefore, we selected the most representative indicators, AP and mAP, as comparative metrics. The results of nine detections are reported in Table 2, where the IoU threshold is set to 0.5.

As depicted in Table 2, our Light-FDD achieves the mAP50 of 0.848, securing the top position among the nine models. The second-best model is YOLOv11n, which is one of the latest lightweight detectors, and attains an mAP50 of 0.836. It is 1.2% lower than our model. For each type of defect, the AP values of our model rank among the top, with Hole, Line, and Stain being the first, and Knot being the third, while other models tend to perform well only on specific types of defects. We also compared our model with another latest lightweight model, MobileNetV4, and found that our mAP is 3.5% higher.

Since precision and recall are typically trade-off metrics, higher precision at higher recall reveals better performance. Hence, the precision-recall (P-R) curve and F1-score were used to illustrate the performance of the detection model. The P-R curve is delineated in Fig. 9, and F1 scores are reported in Table 3. It can be observed from Fig. 9 that the area under the P-R curve of Light-FDD (red curve) is the largest among the comparison models. It indicates that our model achieves the best classification capability and robustness. Furthermore, as shown in Table 3, Light-FDD exhibits the highest F1 score on the average value across all defect categories, with an average score of 0.827, which is 1.5% higher than that of the second-best model. This result indicates that Light-FDD has a significant advantage in detection accuracy. Therefore, it can be concluded that Light-FDD successfully balances detection accuracy and model complexity in fabric defect detection.

Figure 9 P–R curves of different detectors.

The detection results of the aforementioned models on some test sets are shown in Fig. 10. We selected four fabric defect images, which contain all types of defects. For the sake of visualization, we arranged four result images together for display. From the test results, it can be observed that the comparative models exhibit certain limitations. Although the comparative models detected most of the obvious defects, they missed some relatively small ones, and the confidence scores of their detection boxes were also lower than those of our proposed model.

Figure 10 Example detection results of Light-FDD and comparison models.

Ablation experiments

To further validate whether each module of our proposed method is effective, ablution experiments were performed in this subsection. The main improvements of Light-FDD involve the backbone and neck components of the original YOLOv8-n. To systematically analyze the effect of each improvement, we denoted the benchmark model (YOLOv8-n) as A, FasterNet as B, PDCD as C, GCRF attention as D, and VoVDualCSP as E. Our Light-FDD can be represented as A + B + C + D + E, where “+” refers to “introduce”. The evaluation metrics that reflect the detection accuracy were precision, recall, mAP50, and mAP50-95. In addition to the above indicators, several other metrics related to the lightweight model, such as model size, FLOPs (floating-point operations), and FPS (frames per second), are also reported. The results of each defect category are depicted in Table 4 and the overall results are presented in Table 5.

According to the results displayed in Tables 4 and 5, the following conclusions can be drawn: (1) The performance of model A, i.e., the YOLOv8-n, is suboptimal, especially in terms of efficiency. After replacing the backbone part of A with B, the overall mAP50 decreases from 82.4% to 78.9%. The main reason for this accuracy decreases is that B is an ultra-lightweight network, which reduces model complexity and parameter redundancy, making the model faster and more memory-efficient. However, it brings the model has less capacity to capture intricate features. Although the model accuracy declines, the model size and GFLOPs were reduced from 6.4M to 4.6M, and 8.1G to 6.1G, respectively. The FPS increases from 130.4 to 194.8. Therefore, simplifying the backbone is an optimal way to realize a lightweight detection model to realize a lightweight detection because it significantly improves the model’s efficiency in terms of speed and memory usage while maintaining a relatively acceptable level of accuracy.

(2) The parallel dilated convolution is capable of extracting more detailed features, making it better suited for detecting small and indistinct defects. After integrating C (i.e., PDCD block), we observe a significant improvement in accuracy for each defect type and overall performance. Specifically, mAP50 shows a notable increase of 4.6% (from 78.9% to 83.5%), and mAP50-90 rises by 5.7% (from 42.3% to 48.0%). This demonstrates that our downsampling approach effectively enhances feature learning for texture information of the fabric. However, this improvement comes with an increase in computational cost. Specifically, the model size grows from 4.6M to 5.7M parameters, and GFLOPs rises from 6.0 to 6.3, resulting in a decrease in inference speed (FPS drops from 194.8 to 143.3). Although the incorporation of C slightly increases complexity and reduces inference speed, it significantly improves detection accuracy. Compared to the baseline, this model has 1.1% and 0.9% higher mAP50 and mAP50-90, respectively, while maintaining 11% fewer parameters and 22% lower GFLOPs.

(3) From model A + B + C to model A + B + C + D, we can observe that the detection precisions are rising, such as the mAP50 rises by 0.7% (from 83.5% to 84.2%), and the mAP50-95 rises from 48% to 49.1%. However, the computational cost increases slightly, the model size only grows 0.2M, and the GFLOPs increases 0.2G, and the FPS decreases from 143.3 to 135.1. This improvement demonstrates that our designed GCRF attention mechanism is an effective method for enhancing model performance with acceptable computational cost.

(4) Compared with model A + B + C + D, the proposed Light-FDD (i.e., model A + B + C + D + E) achieves a more lightweight design with higher accuracy. Specifically, the model size decreases from 5.9M to 5.5M, the GFLOPs reduce from 6.5G to 6.1G, and the FPS rises from 135.1 to 139.5. In contrast, the model performance was enhanced. For example, the mAP50 improves from 84.2% to 84.8%, and mAP50-95 enhances from 49.1% to 49.7%. This indicates that the E (i.e., VoVDualCSP) module not only reduces the model’s complexity but also improves the detection accuracy.

As shown in Table 5, all proposed modules demonstrate significant contributions to the Light-FDD model. Module B was employed to substantially lighten the model’s backbone, while Modules C and D enhanced detection accuracy with low computational cost. Module E achieves a balanced trade-off between lightweight design and accuracy improvement.

To better illustrate the role of each proposed module in Light-FDD, we employed the CAM heat maps and guided backpropagation method to highlights the key regions of each model iteration. As illustrated in Fig. 11, the heat map clearly shows an order of outcomes for different settings of the models. From left to right, each column is the original images and corresponding heat maps for model A, model A + B, model A + B + C, model A + B + C + D, and model A + B + C + D + E, respectively. Each row displays one type of defects on fabric images. The defects are hole, line, knot, and stain in the top to bottom, respectively. It can be seen that each feature map tends to converge by stage into a target region. The influence of the downsampling operation (i.e. PDCD block) and the attention mechanism (GCRF) is the most significant. As a result, the incorporation of these sophisticated visualizations helps us to gain detailed insights into the performance of Light-FDD modules clearly, and we can conclude that all proposed modules are effective in this article.

Figure 11 Grad-CAM visualization of each module for Light-FDD on different defect images.

The highlighted regions of the network guided by each module closely correspond to the recognized objects.

Comparative experiments on other datasets

To further evaluate the robustness and general applicability of our Light-FDD, the model will be verified in the Smart Diagnosis of Cloth Flaw (SDCF) dataset (Tianchi, 2020). The SDCF is a complex and large-scale dataset for fabric defect research and algorithm evaluation, containing 5,913 fabric defect images. It was collected on-site in a textile production workshop and covered 20 categories of fabric surface defects, such as stain, hole, neps, snag, and so on. However, defect categories are exclusively annotated using serial numbers, so we adopt this manner in this article.

We divided the SDCF into training, validation, and testing sets with a ratio of 7:1:2. This resulted in a training set containing 4,139 defect samples, a validation set with 592 samples, and a testing set comprising 1,182 samples. To illustrate the generalization capability of Light-FDD, we utilized the previously mentioned comparative models and added two large models to conduct experiments for comparison. The results, as detailed in Table 6, indicate that:

Table 6 Results of comparison experiments on the SDCF dataset.

Data	Models	Precision	Recall	mAP50	mAP50-95	Model size (MB)	GFLOPs	FPS	
Val	ShuffleNetv2	0.395	0.317	0.302	0.133	5.9	7.4	102.9	
GhostNetv2	0.46	0.348	0.343	0.158	13.3	8.7	46.32	
MobileNetv3-S	0.418	0.334	0.336	0.155	5.0	5.4	97.2	
MobileNetv4-S	0.37	0.33	0.303	0.137	8.5	8.0	126.4	
YOLOv5n	0.465	0.351	0.344	0.151	5.3	7.1	169.1	
YOLOv8n	0.42	0.306	0.298	0.125	6.3	8.1	157.5	
YOLOv10n	0.35	0.3	0.26	0.12	5.8	8.3	109.5	
YOLOv11n	0.455	0.337	0.336	0.158	5.5	6.5	116.6	
REDETR-l	0.281	0.276	0.161	0.0742	63.2	103.5	10.1	
SwinTransformer	0.355	0.258	0.251	0.109	7.6	9.7	29.2	
Light-FDD (ours)	0.532	0.357	0.392	0.182	5.6	6.1	124.5	
Test	ShuffleNetv2	0.47	0.327	0.324	0.146	5.9	7.4	102.9	
GhostNetv2	0.435	0.379	0.344	0.163	13.3	8.7	46.32	
MobileNetv3-S	0.397	0.339	0.325	0.153	5.0	5.4	97.2	
MobileNetv4-S	0.394	0.339	0.313	0.14	8.5	8.0	126.4	
	YOLOv5n	0.44	0.344	0.334	0.15	5.3	7.1	169.1	
YOLOv8n	0.372	0.317	0.297	0.131	6.3	8.1	157.5	
YOLOv10n	0.413	0.337	0.312	0.15	5.8	8.3	109.5	
YOLOv11n	0.473	0.342	0.341	0.158	5.5	6.5	116.6	
REDETR-l	0.267	0.249	0.152	0.064	63.2	103.5	10.1	
SwinTransformer	0.346	0.288	0.256	0.106	7.6	9.7	29.2	
Light-FDD (ours)	0.496	0.386	0.388	0.183	5.6	6.1	124.5	
Note:

Bold values indicate the best values and underlined values denote the second-best ones.

In terms of detection accuracy, our proposed Light-FDD has obtained the best results on validation and testing sets of SDCF datasets than other detection algorithms. Specifically, on the validation set, the precision, recall, mAP50, and mAP50-95 of Light-FDD reach 0.532, 0.357, 0.392, and 0.182, respectively. On the testing set, these values are 0.496, 0.386, 0.388, and 0.183, respectively. Notably, the mAP50 of our model is 4.8% and 4.4% higher than the second-best model on validation sets and testing sets, respectively. To further validate the credibility of the superior performance of the proposed model, we also conducted a comparative experiment involving two larger models, i.e., RT-DETR (Zhao et al., 2024) and SwinTransformer (Liu et al., 2021). The comparative results reaffirmed that Light-FDD consistently outperformed these larger models in detection accuracy. Therefore, we can conclude that Light-FDD has high detection accuracy on other types of datasets, indicating that our proposed methods have strong generalization ability.

In additional, from the perspective of model lightweighting, Light-FDD also exhibits significant advantages. Its model size and GFLOPs are notably reduced compared to the baseline (i.e., YOLOV8n). Although there is a minor decrease in inference speed, the maintained speed of 124.9 FPS remains adequate for real-time detection applications. Therefore, it can be concluded that Light-FDD is an effective lightweight model with high detection accuracy and low complexity.

Conclusions

In this article, we proposed a lightweight fabric defect detection, Light-FDD, which achieves high detection accuracy with low mode complexity. It utilized an improved FasterNet as the backbone, which not only reduces computational costs but also minimizes memory access, making the network as fast as possible. To enhance detection performance, we introduced the parallel dilated convolution downsampling (PDCD) block, which jointly extracts local and contextual features more precisely. We also presented the global context and receptive-field (GCRF) attention mechanism to make the model focus on key regions. Additionally, a lightweight CSP layer, VoVDualCSP, was designed by dual convolution for the neck network. It enhances the multi-scale feature fusion capability of Light-FDD with a lower computational complexity. Experimental results demonstrated that Light-FDD outperforms the other lightweight models for fabric defect detection tasks on public datasets. Although Light-FDD exhibits high detection performance with fewer model parameters, defects similar to fabric textures, such as knots and lines, have not yet been effectively detected. Therefore, we will focus on enhancing the extraction capabilities of texture and defect features. Furthermore, although Light-FDD was evaluated on two public datasets, its generalization ability still necessitates further experimental validation. To address this, we plan to collect an additional large-scale defect dataset to better meet the requirements of fabric industry applications.

Additional Information and Declarations

Competing Interests

The authors declare that they have no competing interests.

Author Contributions

Zheqing Zhang conceived and designed the experiments, performed the experiments, analyzed the data, performed the computation work, prepared figures and/or tables, authored or reviewed drafts of the article, and approved the final draft.

Kezhong Lu performed the experiments, performed the computation work, authored or reviewed drafts of the article, and approved the final draft.

Gaoming Yang conceived and designed the experiments, analyzed the data, prepared figures and/or tables, authored or reviewed drafts of the article, and approved the final draft.

Data Availability

The following information was supplied regarding data availability:

Data is available at Roboflow: https://universe.roboflow.com/ak03/fabricdefect-zqt9u.

Code is available at GitHub and Zenodo:

https://github.com/czu-zhang/Light-FDD.

czu-zhang. (2025). czu-zhang/Light-FDD: Light-FDD (V1.1). Zenodo. https://doi.org/10.5281/zenodo.16108528.

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
