# Peer review of "A lightweight fabric defect detection with parallel dilated convolution and dual attention mechanism"

_PeerJ Computer Science, doi:10.7717/peerj-cs.3136_

## Round 0.1 · original submission · Major Revisions

The model is only evaluated on a single public dataset, which limits the conclusions about its robustness and general applicability; testing on at least one additional dataset or conducting domain adaptation experiments would provide stronger validation.

The novelty of the proposed modules is not fully convincing, as both the PDCD and GCRF components appear to build on established ideas without clear analytical or comparative justification showing their unique contribution.

The paper lacks a detailed explanation of the training process, including optimizer configuration, batch size, data augmentation strategy, and whether cross-validation or data splits were used consistently.

The ablation study is helpful but does not include enough analysis of performance trade-offs such as speed vs. accuracy or parameter count vs. detection quality; these discussions are important for deployment-oriented models.

Reviewer 2 ·

Basic reporting

This paper proposes a lightweight fabric defect detection method based on YOLO-V8. The reduction of the model's parameter number is achieved by introducing a dual attention mechanism and adding a Cross-Stage Partial layer. The writing style of the manuscript is relatively standardized. It is recommended to place the figures and charts in the correct positions in the main text to enhance readability.

Experimental design

1. This article conducted experimental comparisons with several similar methods and also carried out ablation experiments. There are also newer studies, such as
[1] Jun Tang et.al., A lightweight surface defect detection framework combined with dual-domain attention mechanism, https://doi.org/10.1016/j.eswa.2023.121726
[2] Chen, Chang, et al. "LWFDD-YOLO: a lightweight defect detection algorithm based on improved YOLOv8." Textile Research Journal.

2. The last column of the experimental results, in Tables 2 and 3, does not seem to be the average value of the previous columns.

3. In the ablation test, the result of A+B is worse than A. Why?

Validity of the findings

-

Reviewer 3 ·

Basic reporting

Clear and unambiguous, professional English used throughout.

Structure conforms to PeerJ standards, discipline norms, or any deviations are to improve clarity.

The Introduction adequately introduces the subject and makes it clear what the motivation is.

Experimental design

The article content is within the Aims and Scope of the journal and article type.

Rigorous investigation performed to a high technical & ethical standard.

Methods described with sufficient detail & information to replicate (code, dataset, computing infrastructure, reproduction script)

There is a discussion on data preprocessing, and it is sufficient.

The evaluation methods, assessment metrics, and model selection methods are adequately described.

Validity of the findings

Meaningful replication is encouraged where rationale & benefit to the field are clearly stated.

Conclusions are well stated & limited to supporting results.

The experiments and evaluations have been performed satisfactorily.

There is a well-developed and supported argument that meets the goals set out in the Introduction.

The Conclusion identifies unresolved questions/limitations/future directions.

Additional comments

It looks interesting and can provide some sort of vision to readers.

Although the introduction and background provide context, and the literature is well referenced and relevant, some other studies have been conducted in this field.

Reviewer 4 ·

Basic reporting

This paper introduces a lightweight convolutional neural network model, named Light-FDD, designed for real-time fabric defect detection in industrial settings. The proposed architecture integrates a parallel dilated convolution downsampling block (PDCD) to maintain receptive field coverage while reducing computational cost, and a dual attention module (GCRF) to enhance feature representation by focusing on critical regions. The method is evaluated on a publicly available fabric defect dataset and benchmarked against several lightweight detection models. The results show improved mean Average Precision (mAP) and reduced parameters, supporting the suitability of Light-FDD for edge computing scenarios. While the proposed method is practically relevant and the design is intuitively motivated, the paper requires substantial revision to improve clarity, experimental rigor, and methodological justification before it can be considered for publication.

The model is only evaluated on a single public dataset, which limits the conclusions about its robustness and general applicability; testing on at least one additional dataset or conducting domain adaptation experiments would provide stronger validation.

The novelty of the proposed modules is not fully convincing, as both the PDCD and GCRF components appear to build on established ideas without clear analytical or comparative justification showing their unique contribution.

The paper lacks a detailed explanation of the training process, including optimizer configuration, batch size, data augmentation strategy, and whether cross-validation or data splits were used consistently.

The ablation study is helpful but does not include enough analysis of performance trade-offs such as speed vs. accuracy or parameter count vs. detection quality; these discussions are important for deployment-oriented models.

Experimental design

The attention mechanism is introduced as a key part of the design, yet the explanation is brief and lacks intuitive or visual analysis showing how it improves the feature maps or localization accuracy.

Several terms and notations used in the architecture description are introduced without sufficient clarity, making it difficult for readers to follow the technical flow, particularly in the equations and module definitions.

The comparative experiments are limited to lightweight detectors and do not include baseline models with higher accuracy but higher computational cost, which would help contextualize the trade-offs made in this work.

Validity of the findings

Language quality and formatting need revision throughout the paper, as there are noticeable grammatical errors, inconsistent formatting of equations and tables, and unclear figure captions that reduce the overall readability.

The literature citation is not adequate, and related work on machine learning should be discussed.

---

## Round 0.2 · accepted · Accept

I agree that the author has adequately addressed the concerns raised by previous reviewers. The paper is well-structured, clearly written, and presents reliable results.

Reviewer 4 ·

Basic reporting

The author has adequately addressed the concerns raised by previous reviewers. The paper is well-structured, clearly written, and presents reliable results. It meets the necessary standards for publication

Experimental design

The author has adequately addressed the concerns raised by previous reviewers. The paper is well-structured, clearly written, and presents reliable results. It meets the necessary standards for publication

Validity of the findings

The author has adequately addressed the concerns raised by previous reviewers. The paper is well-structured, clearly written, and presents reliable results. It meets the necessary standards for publication

Additional comments

The author has adequately addressed the concerns raised by previous reviewers. The paper is well-structured, clearly written, and presents reliable results. It meets the necessary standards for publication